Mak *et al*. eLife 2015;4:e07178. DOI: 10.7554/eLife.07178

# Characterization of the finch embryo supports evolutionary conservation of the naive stage of development in amniotes

Siu-Shan Mak[1], Cantas Alev[2†], Hiroki Nagai[2†], Anna Wrabel[1,2†], Yoko Matsuoka[1], Akira Honda[1], Guojun Sheng[2]*, Raj K Ladher[1,3]*

[1]Laboratory for Sensory Development, RIKEN Center for Developmental Biology, Kobe, Japan; [2]Laboratory for Early Embryogenesis, RIKEN Center for Developmental Biology, Kobe, Japan; [3]National Center for Biological Sciences, Bengaluru, India

**Abstract** Innate pluripotency of mouse embryos transits from naive to primed state as the inner cell mass differentiates into epiblast. In vitro, their counterparts are embryonic (ESCs) and epiblast stem cells (EpiSCs), respectively. Activation of the FGF signaling cascade results in mouse ESCs differentiating into mEpiSCs, indicative of its requirement in the shift between these states. However, only mouse ESCs correspond to the naive state; ESCs from other mammals and from chick show primed state characteristics. Thus, the significance of the naive state is unclear. In this study, we use zebra finch as a model for comparative ESC studies. The finch blastoderm has mESC-like properties, while chick blastoderm exhibits EpiSC features. In the absence of FGF signaling, finch cells retained expression of pluripotent markers, which were lost in cells from chick or aged finch epiblasts. Our data suggest that the naive state of pluripotency is evolutionarily conserved among amniotes.

*For correspondence: sheng@cdb.riken.jp (GS); rajladher@ncbs.res.in (RKL)

†These authors contributed equally to this work

Competing interests: The authors declare that no competing interests exist.

## Introduction

The successful isolation and in vitro culture of embryonic stem cells (ESC) from mouse embryos have enabled technological breakthroughs and revolutionized our understanding of the molecular mechanisms regulating mammalian development (*Evans, 2011*). However, similar applications to other species have been lacking. One conceptual difficulty has been linking the innate pluripotency of the embryo with the characteristics of cultured stem cells, raising the speculation that mESCs may be solely a result of in vitro manipulations (*Pauklin et al., 2011*). In the mouse, ESCs are at naive state (*Nichols and Smith, 2009*), which recent evidence suggests is most similar to cells from embryonic day (E) 4–4.5 of mouse development (*Boroviak et al., 2014*). Cells taken from this stage can give rise to derivatives of all three germ layers as well as germ cells. These cells, both in their native naive state and in culture, express genes associated with pluripotency such as *Oct3/4* (*Pou5f1*), *Sox2*, and *Nanog* (2). Using defined media that includes the presence of either an inhibitor of FGF signaling or its downstream Erk/MAP kinase transduction pathway, mouse ESCs (mESCs) can be propagated while maintaining the expression of these pluripotency markers (*Ying et al., 2008*). A second pluripotent cell type in the mouse, epiblast stem cells (mEpiSCs), is derived from embryos that are later in development (E5.5) and is in what has been termed, the primed state (*Brons et al., 2007*). These cells have a more limited potency and require different culture condition for in vitro propagation (*Lanner and Rossant, 2010*), with a dependency on FGF-mediated ERK activation for the maintenance of pluripotent gene expression. Pluripotent ESCs from other mammalian organisms, such as human (*Thomson et al., 1998*; *Schatten et al., 2005*), and from non-mammalian amniotes, such as chick (*Pain et al., 1996*), share this requirement for ERK signaling (*Tesar et al., 2007*). Hence, the primed

**eLife digest** In animals, stem cells divide to give rise to other cells that have specialized roles in the body. 'Pluripotent' stem cells—which are able to produce cells of any type—can be obtained from young mouse embryos. Once grown in the laboratory, these cells are called naive embryonic stem cells (ESCs) and their discovery has been vitally important for understanding how mammals develop. ESCs also have considerable medical potential because they could be used to repair or replace tissues that have been lost to injury or disease.

A family of proteins called fibroblast growth factors (FGFs) triggers naive ESCs to mature into another class of stem cell that are 'primed' to only produce particular types of cells. Curiously, the stem cells that have been collected from other mammal embryos are already in this primed state. Therefore, biologists wonder whether the naive state is exclusive to mice embryos, or whether it is present in other animals but has so far remained undetected.

The development of chick and other bird embryos shares many parallels with that of mammals. However, embryos in chicken eggs do not contain naive ESCs. It is possible that this is due to chicken eggs being laid when the embryos have reached a later stage in development where the naive stem cells have already matured into the primed cells. Here, Mak et al. compared the stem cells in chick embryos to those from another bird called the zebra finch.

The experiments show that the finch embryos contain stem cells that share several features with mouse ESCs. In particular, these finch cells continue to express genes that are required for the naive state to be maintained in the absence of FGF. On the other hand, these genes are switched off in cells from chick embryos and in older zebra finch stem cells.

Mak et al.'s findings show that finch eggs are laid at an earlier stage of embryo development than chicken eggs. The experiments also suggest that both birds and mammals have naive pluripotent stem cells during the early stages of embryo development. In future, the zebra finch could be used as a model to study stem cells and other aspects of animal development.

state of pluripotency is evolutionarily conserved in mammalian and non-mammalian amniotes. However, the naive state has so far only been confirmed in the mouse (*Ying et al., 2008*) and rat (*Buehr et al., 2008*; *Li et al., 2008*; *Chen et al., 2013b*), raising the possibility that this state is not conserved among the amniotes. More recent reports suggested that with specific reprogramming factors and culture conditions such a naive state may also exist for human ESCs, although the exact nature of these naive-type human cells is under debate (*Takashima et al., 2014*; *Theunissen et al., 2014*; *Ware et al., 2014*). Identifying the naive state of embryogenesis in other species is therefore central to our conceptual understanding of pluripotent stem cells.

A comparative embryology approach to address this question should include non-mammalian amniotes. These include avian species, which share key molecular and cellular features of epiblast morphogenesis with the mammals (*Sheng, 2014*), yet are evolutionarily distant enough to serve as an outgroup. As in all amniotes, fertilization of avian oocytes takes place internally and avian embryos undergo some development prior to egg-laying (oviposition). The most widely used avian developmental models are chicken (*Gallus gallus*), quail (*Coturnix japonica*), and zebra finch (*Taeniopygia guttata*). However, chicken embryos at oviposition are already at a late blastula/early gastrulation stage (Eyal-Giladi and Kochav (EGK) stage X) (*Eyal-Giladi and Kochav, 1976*). These embryos have a columnar epithelialized epiblast overlying scattered hypoblasts and are thus morphologically similar to E5.5 mouse embryo, later than the stage at which mESCs can be derived. The Japanese quail embryos are laid at a stage later than the chicken embryos (*Sellier et al., 2006*), while the ovipositional stage of zebra finch embryos is unclear. Early-staged avian embryos can generate cells that show some of the characteristics of mammalian ES cells (*Jean et al., 2015*) after the introduction of reprogramming factors (*Rossello et al., 2013*; *Dai et al., 2014*) or after the manipulation of culture conditions (*Pain et al., 1996*; *Jean et al., 2013*). However, it was not clear how the pluripotency generated be exogenous factors related to the pluripotent state of cells in the embryo. We decided to investigate the early development of the zebra finch (hereafter referred to as the finch) in more detail. The finches are a model system commonly used in neurobiological studies of social behavior (*Brazas and Shimizu, 2002*; *Svec et al., 2009*), vocalization, and learning (*Jarvis, 2004*; *Petkov and Jarvis, 2012*). These studies have led to an

increased focus on the developmental neurobiology of zebra finch (*Charvet and Striedter, 2009*; *Chen et al., 2013a*), and in turn the embryology of the finch (*Murray et al., 2013*). In addition, the finch genome has been sequenced (*Warren et al., 2010*) and their relatively small adult body sizes (4–7 times smaller than adult chickens) and shorter generation time (2–3 months for finches vs 4–6 months for the chickens) makes it feasible to breed them within a normal laboratory setting. Thus, we asked whether finch embryos could be used for early embryogenesis and ESCs studies, complementing existing studies (*Rossello et al., 2013*), which together may lead to potential technological breakthroughs that facilitate genetic and functional investigations in this model organism.

Here, we report the first molecular characterization of ovipositional finch blastoderms by quantitative RT-PCR, immunohistochemical staining, and in situ hybridization. Our results suggested that finch embryos are laid at stage EGK-VI to EGK-VIII, much earlier than the chicken embryos, and are morphologically more similar to the E4–E4.5 mouse embryos from which mESCs can be derived. Cells derived from the finch blastoderm at oviposition and cultured in the presence of a MEK inhibitor and Leukemia inhibitory factor (LIF) retained expression of the pluripotent markers; *Nanog, PouV* expression, and alkaline phosphatase (AP) activity. In contrast, chicken cells taken from newly laid embryos and cultured under the same conditions did not produce *Nanog, PouV*, or AP-positive aggregates. Our data suggest that birds and mammals share a common regulatory mechanism in the maintenance of pluripotency. Finch embryos are ideally suited for the establishment and characterization of avian ESCs, and the incorporation of recent technical improvements (*Dai et al., 2014*) could lead to the finch becoming a tractable avian model for genetics and regenerative medicine.

## Results

### Finch oviposition is at EGK-VI prior to subgerminal cavity expansion

Avian embryos undergo varying degrees of intrauterine development prior to oviposition (egg-laying). Chick oviposition is at the late blastula/early gastrula stage (EGK-X to EGK-XI). In other *Galloanserae*, oviposition ranges from EGK-VII (Turkey [*Gupta and Bakst, 1993*] and Duck [*Sellier et al., 2006*]) to EGK-XI (the quail [*Stepinska and Olszanska, 1983*]). Ratite embryos are laid at EGK-X, similar to the chick embryos (*Nagai et al., 2011*). The oviposition stage of *Passerine* (songbird) species has not been carefully investigated, although gross morphology of newly laid embryos of the zebra finch and society finch (Bengalese finch) suggested that they are younger than EGK-X (*Yamasaki and Tonosaki, 1988*; *Agate et al., 2009*; *Murray et al., 2013*). Due to the difficulty in retrieving pre-ovipositional (<EGK-X) chick embryos (*Nagai et al., 2015*), we decided to further explore the potential advantages of finch eggs and carried out a detailed morphological and molecular characterization of ovipositional finch embryos in order to evaluate their potential use for comparative stem cell studies as well as for genome engineering.

Finch eggs are laid in intervals of between 20 and 29 hr, with most laid following a 24- to 25-hr periodicity. Based on the EGK staging system for pre-oviposition chick embryos (*Eyal-Giladi and Kochav, 1976*), we found that finch embryos from newly laid eggs ranged from EGK-VI to EGK-VIII, with ~10% of eggs at EGK-VI and the rest at EGK-VII and -VIII. To probe morphological and molecular heterogeneity of the finch blastoderm, we first used section analysis of embryos stained for Hhex, a marker for endoderm cells (*Thomas et al., 1998*) (*Figure 1A–C*). At EGK-VI (early blastula), the embryo was composed of similar-sized blastoderm cells organized into 7–8 layers (*Figure 1A*). The subgerminal (blastocoel) cavity was not as apparent as in similarly staged chick embryos (*Eyal-Giladi and Kochav, 1976*; *Sheng, 2014*). By EGK-VIII (mid-blastula stage) (*Figure 1B*), the subgerminal cavity and the area pellucida became apparent and the embryo was thinner at the center of the blastoderm (4–5 cell-thick) than at EGK-VI. Cells located apically, closer to the vitelline membrane (the putative epiblast precursors) were smaller than those located closer to the yolk side (the putative hypoblast precursors). A low level of *Hhex* expression was detected in the hypoblast precursors. Post-ovipositional incubation of around 6 hr yielded finch embryos at EGK-X (*Figure 1C*). At this stage, the epiblast and hypoblast were clearly separated and the epiblast showed a columnar epithelial organization. In addition, strong *Hhex* expression was detected in the hypoblasts.

Epithelial organization is typically maintained by adherens junctions mediated extracellularly by homophilic interactions of E-cadherin molecules and intracellularly by adhesion complex components such as ß-catenin (*Takeichi, 2014*). To understand the transformation of the epithelial morphology between EGK-VI to EGK-X, we asked if there were changes to E-cadherin and ß-catenin localization

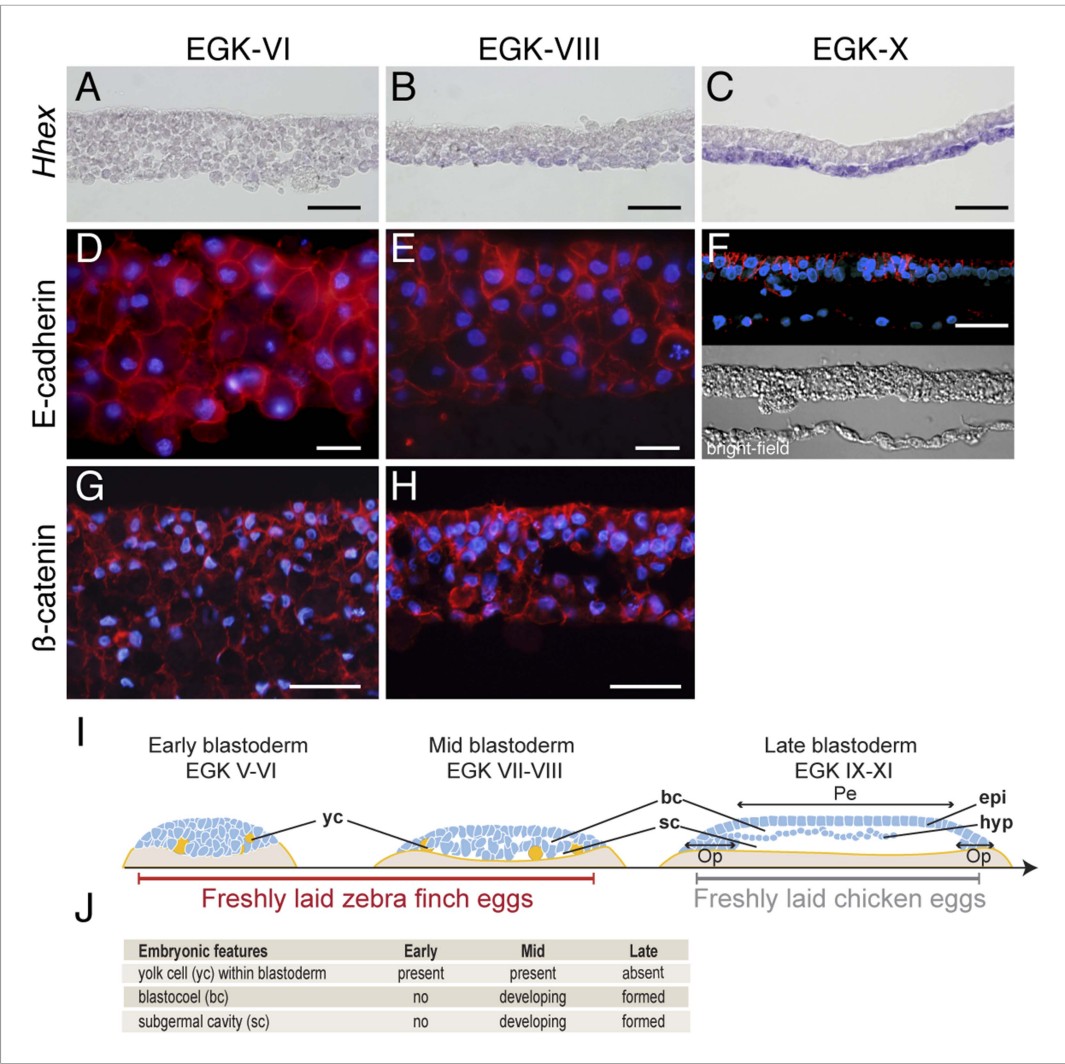

Figure 1. Morphology of the finch blastoderm at oviposition. (**A–C**) Finch embryos were sectioned after *Hhex* in situ hybridization to reveal the morphology of the blastoderm. At EGK-VI the blastoderm is thick, with no sign of epithelialization, and does not express *Hhex* (**A**). The blastoderm is thinner at EGK-VIII but still has not overt epithelialisation, and express low level of *Hhex* transcripts in the cells of the yolk side (**B**). *Hhex* expression can be detected clearly at EGK-X and is confined to a morphologically distinct hypoblast layer (**C**). (**D–F**) Staining of E-cadherin reveals the extent of epithelialization of the finch blastoderm. At EGK-VI, most of the blastodermal cells appear unpolarized, however, express E-cadherin (**D**). At EGK-VII, while thinner, the blastoderm does not show any columnar cells typical of epithelial organization (**E**). At EGK-X, clear epithelial organization can be visualized, with polarized E-cadherin and morphological segregation in the bright-field image (**F**). (**G**, **H**) ß-catenin staining confirms the lack of epithetialisation at finch oviposition stages. The pattern at EGK-VI is similar to E-cadherin staining (**G**) as is the pattern at EGK-VIII (**H**). In **A–G**, scale bar represents 100 μm. (**I**) Schematic of the morphology of the early, mid, and late finch blastula: The late finch blastoderm is similar to the chick embryo at oviposition. Yc—yolk cells; bc—blastocoel; sc—subgerminal cavity; epi—epiblast; hyp—hypoblast; Op—area opaca; Pe—area pellucida. (**J**) Table below showing the features of the finch at early stages.

consistent with epithelialization of the early embryo. At EGK-VI, both E-cadherin (*Figure 1D*) and ß-catenin (*Figure 1G*) were expressed in all cells. Staining did not appear to be enriched in any particular region of the cell and epithelialization was not apparent. By EGK-VIII, formation of an epithelial cell-sheet was not observed although the nuclei of the outermost, putative epiblast, cells appeared more regularly organized (*Figure 1E,H*). In contrast, deep, putative hypoblast, cells appeared to down-regulate both proteins. By EGK-X, E-cadherin was up-regulated in the lateral regions of the epiblast cells

(*Figure 1F*). Their epithelial morphology was further evidenced by formation of a discrete columnar cell-sheet. Analysis of chick embryos at oviposition revealed that it was more similar to the finch embryo after incubation (*Figure 1I*), sharing features with the finch late blastula (*Figure 1J*).

## Developmental lineages are molecularly segregated at EGK-VI prior to epithelialization of the epiblast

The reorganization of the blastoderm into an epithelialized epiblast and hypoblast raised the possibility that epiblast formation may be concomitant with acquisition of cell fate. The early blastoderm consists of cells fated to become the epiblast and hypoblast as well as germ cells (*Eyal-Giladi et al., 1981*; *Hatada and Stern, 1994*; *Lawson and Schoenwolf, 2003*). We thus asked whether these cell identities have already been established in the finch embryo at oviposition. As oviposition ranged from EGK-VI to EGK-VIII, we examined these stages.

In chick, the germ cell fate is marked by immunoreactivity to the stage-specific embryonic antigen 1 (SSEA-1) and epithelial membrane antigen 1 (EMA-1) antibodies (*Jung et al., 2005*). In finch embryos, immunoreactivity of these proteins were observed predominantly in the epiblast precursor (*Figure 2A,B,E–H*), with SSEA-1 staining being stronger than EMA-1. Few hypoblast precursor cells were positive for these markers (*Figure 2C,D,E–H*). In mouse embryos, Gata4 expression marks the specification of the hypoblast/primitive endoderm (*Soudais et al., 1995*) and the SoxB family member, Sox2 expression marks the epiblast precursors (*Avilion et al., 2003*). In chick, a different SoxB family member, Sox3 is expressed in epiblast precursors (*Rex et al., 1997*). We thus used Sox3 and Gata4 to determine the presence and distribution of these cell types in EGK-VI/VIII finch embryos. Both immunostaining and in situ hybridization confirmed that Sox3 and Gata4 could be detected at finch oviposition and that lineage segregation between these two cell types could be distinguished

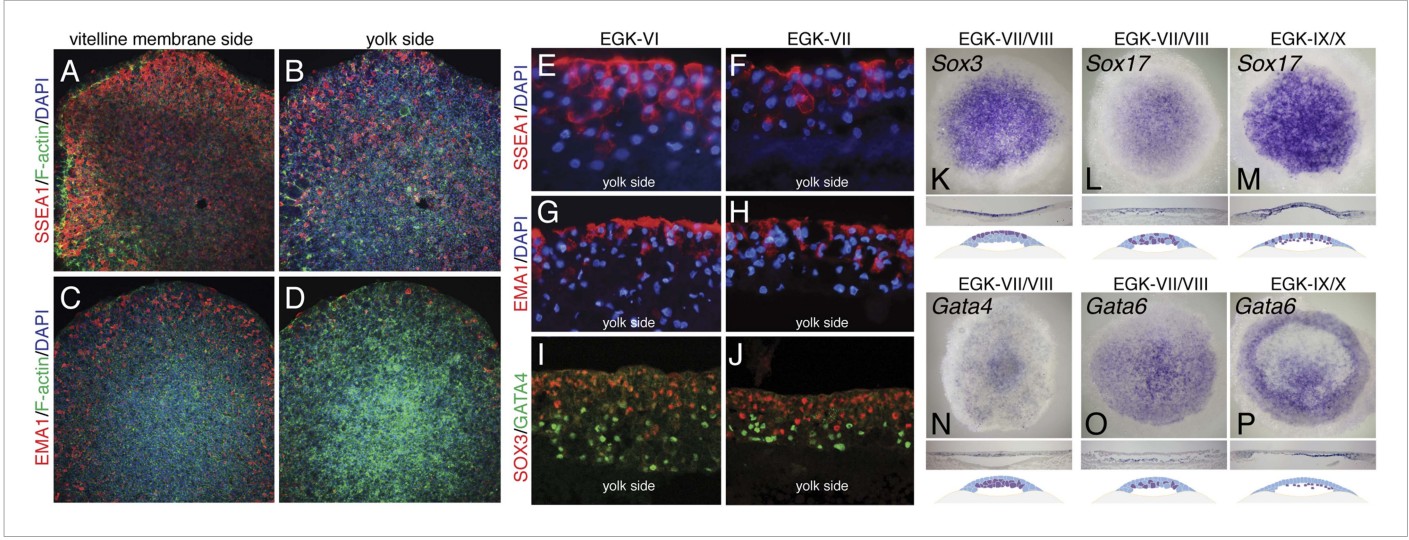

**Figure 2**. Molecular characterization of the finch blastoderm at oviposition. (**A–D**) Whole-mount staining of EGK-VIII finch embryos of the distribution of putative germ cell markers SSEA-1 (**A**, **B**) and epithelial membrane antigen 1 (EMA-1) (**C**, **D**) showing views from the vitelline membrane side (**A**, **C**) and yolk (lower) side (**B**, **D**). (**E**, **F**) Sections of finch embryos at oviposition showing the distribution of SSEA-1 positive cells at EGK-VI (**E**) and EGK-VIII (**F**) to predominantly the upper side of the blastoderm. (**G**, **H**) Sections of finch embryos at oviposition showing the distribution of EMA-1 positive cells at EGK-VI (**G**) and EGK-VIII (**H**) to predominantly the upper side of the blastoderm. (**I**, **J**) Sections of finch embryos at oviposition showing the distribution of SOX3 and GATA4 protein at EGK-VI (**I**) and EGK-VIII (**J**) SOX3 and GATA4 are localized to opposite sides of the blastoderm. (**K**) In situ hybridization of *Sox3* at oviposition in the finch blastoderm. Section analysis, summarized in the cartoon, shows transcripts are present throughout the upper layer of the blastoderm. (**L**, **M**) In situ hybridization of *Sox17* together with section analysis, summarized in the accompanying cartoon, reveals expression dispersed throughout the blastoderm at oviposition (**L**), while at EGK-X expression is more predominant in the hypoblast with scattered epiblast cells (**M**). (**N**) In situ hybridization of *Gata4* at oviposition in the finch blastoderm. Section analysis, summarized in the cartoon, shows transcripts are present throughout the lower layer of the blastoderm. (**O**, **P**) In situ hybridization of *Gata6* together with section analysis, summarized in the accompanying cartoon, reveals expression more predominant in the lower layer/hypoblast at oviposition (**O**) and in the late blastoderm at EGK-IX/X (**P**).

(*Figure 2I–K*). Sox3 was more abundant in the cells at the vitelline membrane side (*Figure 2I,K*) and Gata4 more abundant in the yolk side (*Figure 2J,M*). To verify the specification of the endodermal lineage, three additional markers, *Gata6*, *Sox17*, and *Hhex*, were investigated. In mouse, both Gata6 and Sox17 mark all of endodermal lineages (*Fujikura et al., 2002*; *Kanai-Azuma et al., 2002*). *Sox17* was expressed in the hypoblast in the finch blastoderm from oviposition, with some positive cells also detected in the epiblasts (*Figure 2L*). Expression was similar after epithelization at EGK-IX/X (*Figure 2M*). *Gata6* expression was restricted to the hypoblast lineage (*Figure 2O*) at EGK-VII/VIII, which is more apparent by EGK-X (*Figure 2P*). *Hhex* expression was not detected at oviposition, but endodermal expression could be detected from mid-blastula stage (*Figure 1B,C*). Gene expression at early- and mid-blastula stages was not associated with any overt morphological segregation at either EGK-VI or EGK-VIII. Taken together, these data suggest that the primordial germ cell, epiblast, and hypoblast precursors were already molecularly specified from early-blastula stages, prior to morphological segregation of these cell types.

## Expression of pluripotency-associated markers in finch and chick blastoderm

Our characterization suggested that the finch blastoderm at oviposition was equivalent to the blastocyst stage mouse embryo when the epiblast precursors had not yet polarized to form an epithelium. This is also the stage when mESCs could be derived (*Boroviak et al., 2014*). In mouse embryogenesis, the naive stage is accompanied by the specific expression of a number of pluripotency-associated genes (*Nichols and Smith, 2009*). We thus asked whether markers of pluripotency were also expressed in the finch blastoderm at oviposition. *Nanog* is a transcriptional regulator involved in cell proliferation during early mouse development and in self-renewal of mESCs (*Chambers et al., 2003*; *Mitsui et al., 2003*). However, upon cloning finch *Nanog*, it became apparent that the avian *Nanog* locus has undergone a recent gene duplication event, producing *Nanog* and *Nanog-like* as suggested from chicken data (*Figure 3A*) (*Shin et al., 2011*). Both genes showing a similar level of homology to mouse *Nanog* (*Figure 4B*). Both are expressed in the early EGK-VIII finch blastoderm, predominantly in epiblast cells, but with scattered cells on the yolk side also being positive (*Figure 3C,D*). By EGK-X, *Nanog-like* is differentially enriched in the periphery of the blastoderm, in cells fated for the extra-embryonic tissue (*Figure 3C*). In contrast, *Nanog* is expressed throughout the epiblast of the EGK-X blastoderm (*Figure 3D*). Two additional pluripotency regulators, *PouV* (Pou5F1, the avian homologue of Oct3/4) and *Dnmt3b* (*Calloni et al., 2013*), are also expressed in finch embryos at ovipoisiton in a spatial and temporal pattern similar to that of *Nanog* (*Figure 3E,F*). By EGK-X, expression of these two markers also became restricted to the epiblast.

To gain a broader understanding of the molecular differences between freshly laid finch and chick embryos, we used a comprehensive Q-PCR screen to compare expression levels of a number of genes associated with early embryonic development and pluripotency. These genes have been defined from work in the mammalian species and adapted to avian species more recently. We first tested core pluripotency factors (*Figure 4*). Both finch and chick blastoderms at oviposition showed comparable levels of expression for many of the genes tested, namely *PouV, Nanog, Dnmt3b, Sall4, Cripto, Myc,* and *Sox3*. In contrast, *Nanog-like, Lin28A/B* and *Klf2* were specifically up-regulated in the finch blastoderm. We next looked for expression of genes associated with the naive state of pluripotency. *Fbxo15* and *PRDM14*, which are high in naive state ES cells, are up-regulated in the finch blastoderm. The naive state genes *Tbx3* and *Tfcp2l1* show also higher levels of expression for the finch samples although not as prominent. *Esrrb*, an orphan nuclear receptor that has been used to reprogram fibroblasts into induced pluripotent stem (iPS) cells (*Martello et al., 2013*), was not expressed in either blastoderm at oviposition. Surprisingly, *Nrob1*, another gene associated with the naive state showed reciprocal expression, with high levels detected in the chick blastoderm. Interestingly, expression of above mentioned naive state associated genes could be also detected in chick blastoderm samples, albeit not as robustly as in finch samples.

To clarify this, we tested additional markers to assess whether the finch blastoderm contains features of a full naive state and whether some of these features are lost in the transitional state of the chick blastoderm. The naive state of development includes cells that are able to give rise to the germ cell lineage. We thus analyzed the expression of two putative markers of germ cell fated cells, *Ddx4* and *Dazl*. Both showed higher expression in finch blastoderms at oviposition as compared to chick.

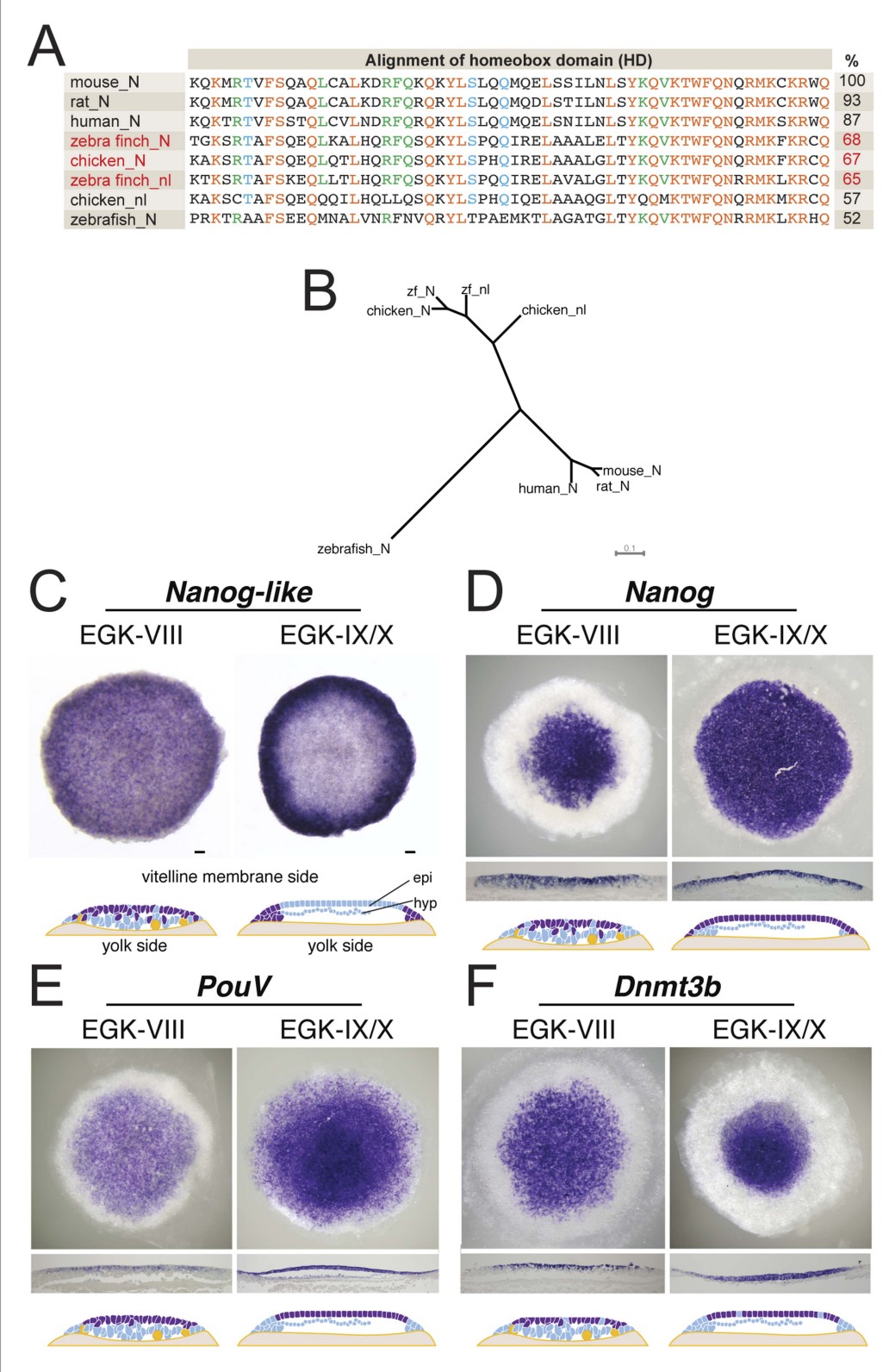

**Figure 3**. Expression of pluripotent markers in the finch embryo at oviposition. (**A**) Comparison of the NANOG (N) and NANOG-LIKE (nl) homeodomains (HD) from chick and the finch compared to the NANOG HD from human, mouse, rat, and zebrafish, together with percentage identity to the mouse NANOG HD. (**B**) Phylogenetic analysis reveals chick and the finch NANOG and NANOG-LIKE are the result of a recent duplication in the avian lineage. *Figure 3. continued on next page*

*Figure 3. Continued*
(**C**) In situ hybridization of *Nanog-like* at oviposition and at EGK-X in the finch blastoderm. Expression is throughout the upper layer of the blastoderm and resolves to peripheral expression in the putative extra-embryonic region. Diagram compiles expression revealed from sections. (**D**) In situ hybridization of *Nanog* reveals expression in the central part of the upper layer at finch oviposition, which expands to cover the entire epiblast at EGK-X. This is shown in sections and detailed in the diagram shown. (**E**) In situ hybridization of the avian *Oct4* homologue *PouV* shows expression throughout the upper layer/epiblast of the EGK-VIII and EGK-X finch embryos. (**F**) Expression of *Dnmt3b* shows expression throughout the upper layer of the blastula at oviposition, which resolves to central epiblast expression at EGK-X.

Cells derived from later, primed, staged blastoderm in mouse are dependent on FGF-mediated Erk activation. We hypothesized that the chick blastoderm at oviposition would show higher levels of FGF. Q-PCR revealed higher levels of *Fgf3*, *Fgf4*, and *Fgf10* in the chick blastoderm as well as slightly elevated levels of *Fgf8* when compared with the finch blastoderm. *Fgf5*, which marks the mature epiblast and is considered to be a mEpiSC marker (*Lanner and Rossant, 2010*), was not detected in either chick or finch blastoderm. LIF mediates Jak-Stat3 pathway activation and has been shown to be essential for the maintenance of ESC pluripotency in the mouse (*Yoshida et al., 1994*). We found that *Lif* and its close homologue *Il6* were both expressed in finch blastoderms but showed lower levels of expression in the chick samples.

## Expression of naive pluripotency-associated markers decreases in older finch blastoderm

Our data suggested that the expression level of naive pluripotency markers is reduced in chick blastoderms with respect to finch blastoderms at oviposition. This could be as a result of the difference in stage between these two embryos, but it is formally possible that this difference results from species differences. To exclude this, we performed Q-PCR on 2 stages of finch embryos; at oviposition and aged to the equivalent of the chick ovipositional stage, EGK-X (*Figure 5*). Markers of general pluripotency, that is, *PouV*, *Nanog*, and *Dnmt3b* show little change between ovipositional finch embryos and finch embryos at EGK-X (*Figure 5A*). Furthermore, *Nanog-like* also showed only slight differences in expression levels between the two stages of finch blastoderm (*Figure 5A*). In contrast, naive pluripotency markers, the expression levels of *Fbxo15*, *Prdm14*, *Tbx3*, and *Nrob1* showed a significant down-regulation in older finch embryos (*Figure 5B*).

## Expression of pluripotency markers is independent of ERK activation in isolated finch blastodermal cells

The molecular characterization described above suggested that the behavior of isolated blastoderm cells from the chick and finch blastoderms at oviposition would be different. We hypothesized that finch blastodermal cells may act more like mESCs, whereas chick blastodermal cells would behave more similarly to mEpiSCs. An important difference between mESCs and mEpiSCs is the effect of extracellular receptor kinase (Erk) pathway inhibition. Naive mESCs culture is aided by blockage of Erk pathway in the presence of LIF (*Nichols and Smith, 2012*), whereas mEpiSCs do not. In order to test this hypothesis, individual blastoderms from finch or chick embryos at oviposition were isolated, dissociated, and cultured. Chick and finch cells that were plated and cultured without LIF and without MAP kinase inhibition formed a monolayer of cells that eventually formed aggregates. However, these aggregated colonies did not show AP activity nor did they express pluripotency-associated genes (*Nanog* or *PouV*). Inclusion of LIF in the culture media had little effect on the morphological and molecular features of either chick or finch-dissociated blastodermal cells. Interestingly, when we added the MEK inhibitor together with LIF in the culture media, finch blastoderm cells formed aggregates that showed strong AP activity, as well as expressed *Nanog* and *PouV* expression (*Figure 6A–C*). This effect was observed either in the presence or absence of GSK3ß inhibition. In contrast, while chick blastodermal cells cultured with LIF and MEK inhibitor did form aggregates, these remained negative for *Nanog* and *PouV* and did not show AP activity (*Figure 6D–F*). We noted that some cells surrounding the chick colonies did show weak AP activity and very weak *PouV* expression, but these were only visible after extended staining incubation times, and were not found in colonies.

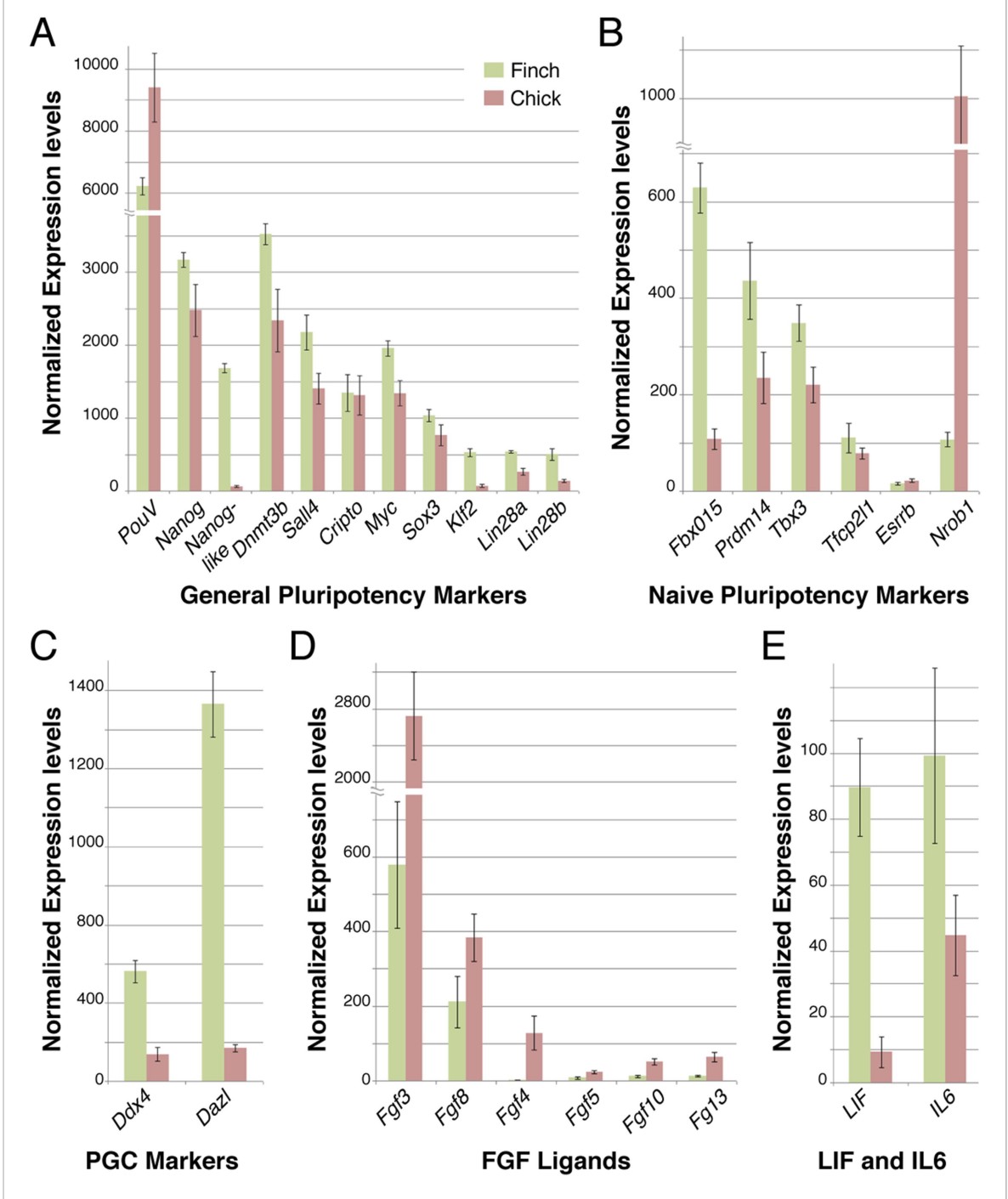

**Figure 4**. Q-PCR characterization of finch and chick blastoderm at oviposition reveals fundamental molecular differences. Quantitative-PCR was used to assess the differences in gene expression between finch (green bars) and chick (red bars) embryos at oviposition. The markers used were associated with general pluripotency (**A**), naive pluripotency (**B**), markers of primordial germ cell development (**C**), fibroblast growth factors (**D**), and leukemia inhibitory factor (LIF) signaling (**E**).

Q-PCR was used to compare the levels of expression of genes associated with pluripotency in the finch blastodermal cells. Three conditions were analyzed; oviposition finch cells with or without MEK inhibitor, and finch blastoderms aged to the equivalent of HH2, dispersed and treated with MEK inhibitor (*Figure 6G*). Maximal expression of the naive pluripotency genes, *Fbxo15* and *Tbx3*, as well

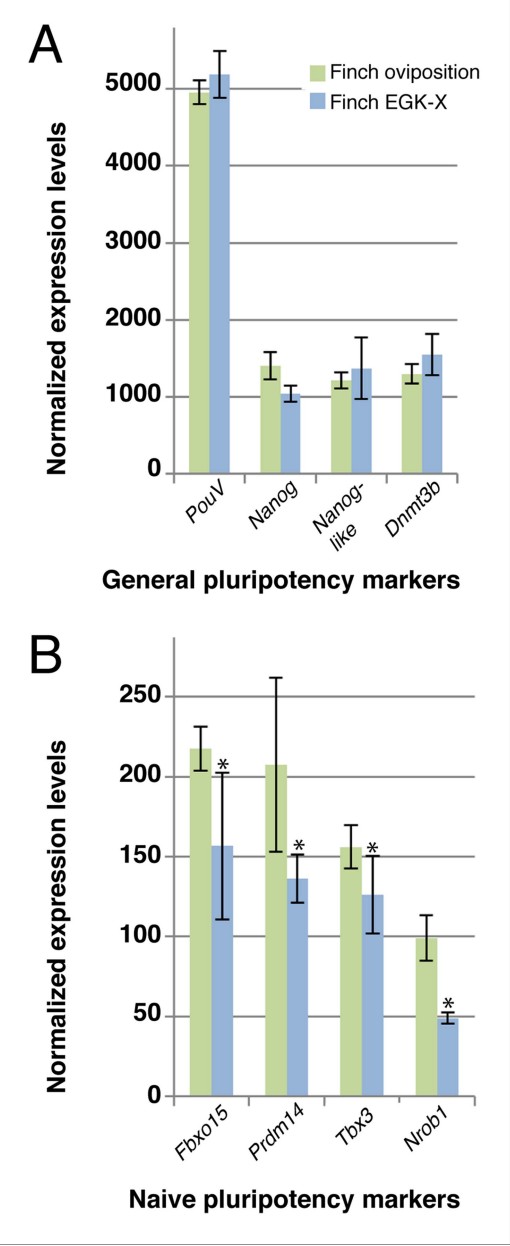

**Figure 5**. Q-PCR characterization of finch blastoderms at oviposition and aged to EGK-X. Quantitative-PCR was used to assess the differences in gene expression between finch blastoderms at laying (green bars) and aged to EGK-X, equivalent to chick oviposition (blue bars). The markers were used to assess general pluripotency (**A**), naive pluripotency (**B**). p-values <0.05 are labeled with an asterisk.

as the general pluripotency marker, *Nanog*, was seen in ovipositional finch blastodermal cells cultured with MEK inhibitor, with less expression seen in cells without MEK inhibitor and in HH2 finch blastodermal cells cultured with MEK inhibitor. These data confirm that pluripotency markers are selectively enriched in early finch blastodermal cells cultured in the presence of MEK inhibitor. *Dnmt3b* showed higher expression in ovipositional finch blastodermal cells without MEK inhibitor. This may indicate a second *Dnmt3b*-expressing cell type that is also enriched in these conditions. In mouse, *Dnmt3b* is initially strongly expressed in the trophectoderm (*Hirasawa and Sasaki, 2009*), and our initial evaluation suggests that some of the enriched cells in our cultures may be extra-embryonic.

## Discussion

The derivation of ES cells must reflect the innate pluripotency of the embryo as well as the ability to maintain this potency in vitro. Mouse data suggest that in permissive strains, ES cells can be derived from embryos at around E4–E4.5 of development, a stage that has been termed the naive stage of embryogenesis (*Nichols and Smith, 2009*; *Boroviak et al., 2014*). The epiblast and hypoblast cell lineage has been specified at this stage and is dispersed throughout the inner cell mass; however, epithelialization has yet to occur (*Plusa et al., 2008*; *Artus et al., 2011*). We find that the finch embryo at oviposition displays similar characteristics: although the epiblast, hypoblast, and germ line are specified, these cells are dispersed throughout the multi-layered blastoderm, which has yet to epithelialize. It is likely that the sorting mechanisms that ensure segregation of germs layers are similar to those in the mouse, but would require further study. However, given the accessibility of early avian development in the finch embryo, these and other comparative studies to test hypotheses on the evolution of germ layer formation in amniotes are possible.

Pluripotent gene expression in blastodermal cells derived from finch embryos is independent of the Erk pathway. This is also an important feature of mouse ES cells. In contrast, in our culture regimen, chick blastodermal cells do not

retain pluripotent gene expression when cultured with an ERK pathway inhibitor, a feature of mEpiSC. Our data thus suggest that the finch blastoderm at oviposition is likely to correspond to the naive stage of embryogenesis. Furthermore, the evolutionary distance between finch and mouse suggests that the naive stage during embryogenesis is conserved amongst amniotes and that this stage likely corresponds to the pre-epithelialized epiblast in amniotes (*Figure 7*).

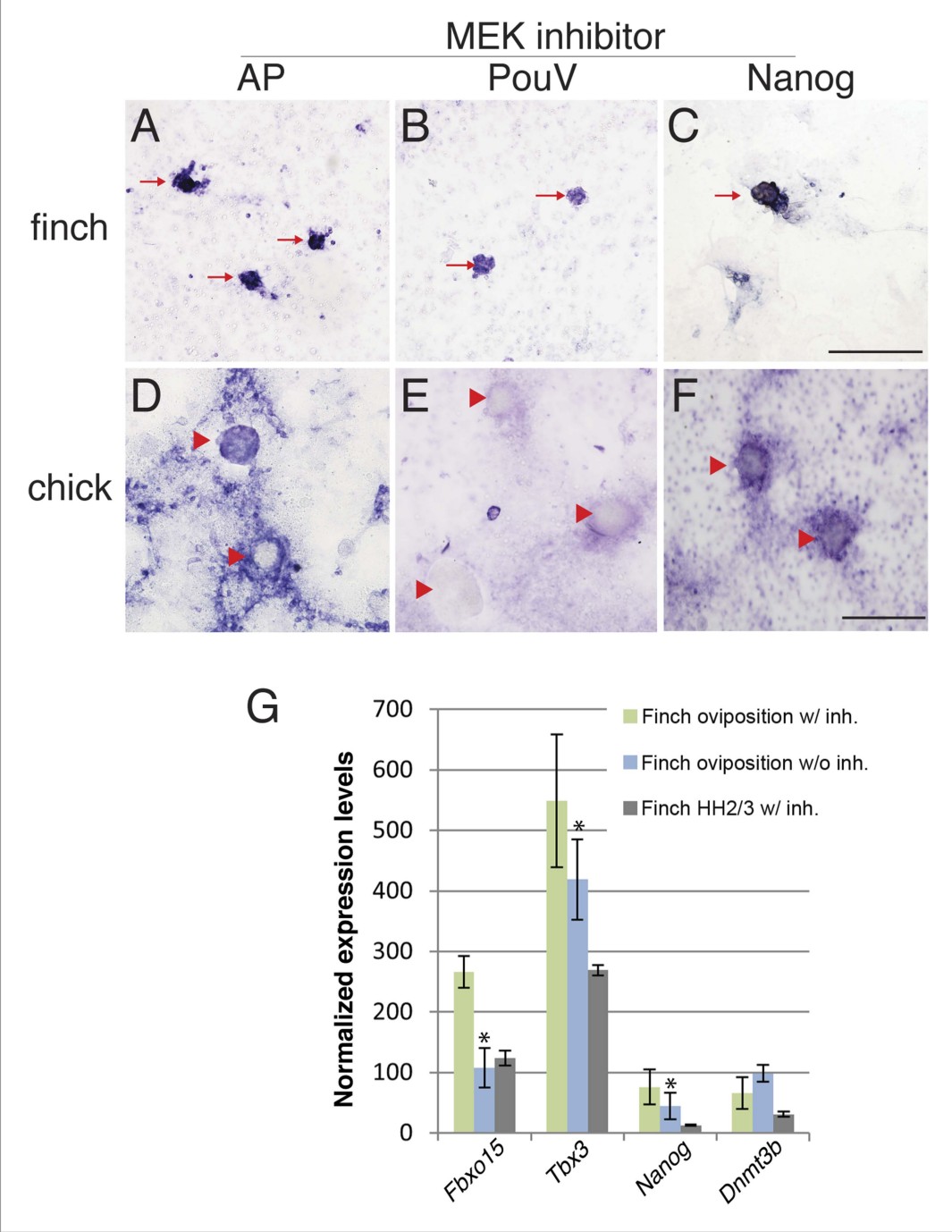

**Figure 6**. Finch blastodermal cell cultures retain markers of pluripotency even in the presence of MEK inhibitor. Finch and chick embryos at oviposition were dissociated and cultured in the presence of LIF and the MEK inhibitor, PD0325901, for 4 days. (A–C) Finch blastodermal cells form aggregates (red arrows) that show alkaline phosphatase (AP) activity (A) as well as expression of *PouV* (B) and *Nanog* (C). (D–F) Chick blastodermal cells formed aggregates (indicated as red arrowheads), but these did not show AP activity (D) nor express *PouV* (E) and *Nanog* (F) even after extended periods of staining. (G) Q-PCR analysis of finch ovipositional blastoderms cultured in the presence of LIF and the MEK inhibitor, PD0325901, for 4 days, as well as finch blastoderm aged to HH2/3 cultured in the presence of LIF and PD0325901. Marker analysis was used to ascertain the pluripotent state of cells in each culture condition. T-tests were used to determine the significance of the difference in markers gene expression in finch ovipositional blastodermal cell culture with or without MEK inhibition. p-values <0.05 are labeled with an asterisk.

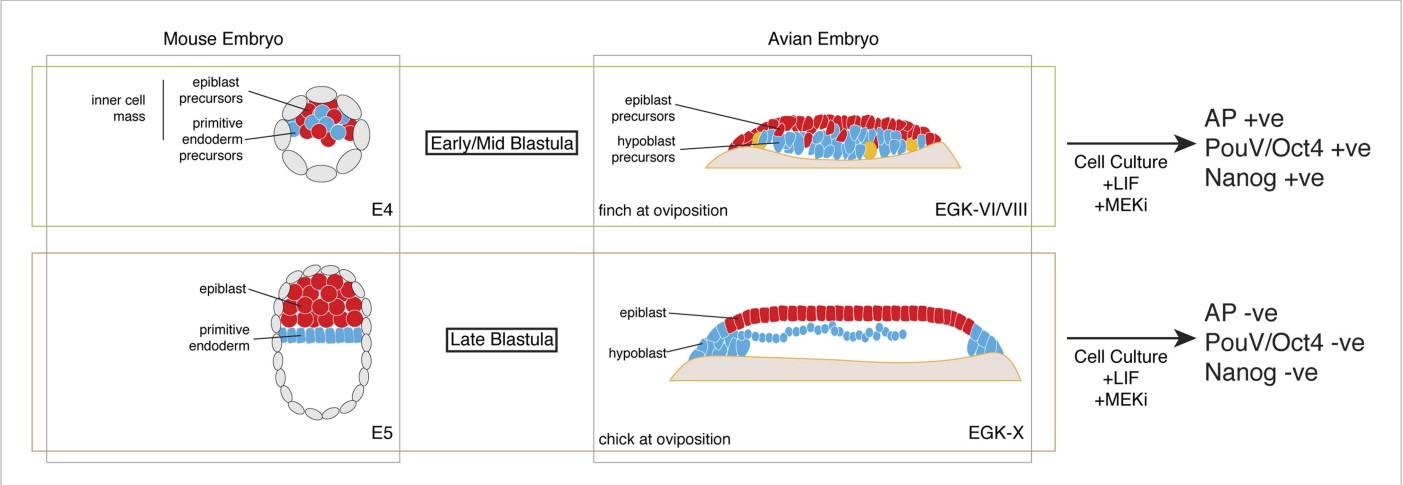

**Figure 7**. Putative conservation of the naive stage during amniote embryogenesis. Similar to mouse pre-implantation embryos (around E4), prior to epiblast epithelialisation, zebra finch embryos at oviposition are at early–mid-blastula stages (EGK-VI/VIII) show no overt morphological segregation despite the expression of epiblast and primitive endoderm/hypoblast markers in distinct sub-populations of cells with the embryo. Cells from mouse and finch at this stage do not require ERK activation for the maintenance of pluripotent marker expression in vitro. Chick embryos at oviposition are more similar to the mouse blastula where epithelialisation has occurred. Similarly, cells from chick and mouse embryos at this stage are unable to maintain pluripotent marker expression when ERK signaling is inhibited.

The relationship between the pre-epithelialized epiblast and pluripotency is not clear. Recent data suggest that a mesenchymal to epithelial transition is important for efficient reprogramming of somatic cells into iPScells (*Li et al., 2010*; *Samavarchi-Tehrani et al., 2010*). Furthermore, E-cadherin, which is important in establishing epithelia, mediates LIF signaling during ES cell self-renewal (*del Valle et al., 2013*). While, this does seem contrary to the notion that ES cell derivation occurs from the pre-epithelialized epiblast, we found that despite a lack of epithelialization of the newly laid finch blastoderm, E-cadherin is still highly expressed. This is similar to E-cadherin expression profile in the mouse late blastocyst (*Thomas et al., 2004*). Thus, more than a mesenchymal to epithelial transition, perhaps the expression of E-cadherin represents a metastable, transitional state during mesenchymal–epithelial transition that is permissive for ES cell derivation. This also suggests that the ability to form epithelial-type adhesive contacts is present but inhibited at early/mid-blastula stages. This may be due to the presence of endogenous inhibitors of E-cadherin homophillic interactions, or alternatively it may be simply that the full repertoire of epithelial-type adhesion molecules is not yet expressed. Consistent with this is the differing adhesive properties of finch and chick blastodermal cells in culture. Chick blastodermal cells were able to adhere to a wide variety of substrates; however, optimal growth of finch blastodermal cells was only observed when type IV collagen was used as a substrate.

The question then remains: how does the pluripotent characteristics of isolated cells relate to the epiblast? One scenario could be that the epiblast consists of mixed populations of cells in varying states of pluripotency, and that the culture conditions employed can stabilize, select for, induce, or enrich for a particular state. Several lines of evidence suggest that this is the case. The use of MEK inhibition promotes the production of ES cells from the epithelialized epiblast of mouse strains that are otherwise refractory for ES cell derivation. Conversely, EpiSC cells can be isolated from the pre-epithelialized epiblast of the majority of mouse strains that do not readily produce ES cells (*Najm et al., 2011*). This suggests a mixed population of cell states, and our data support this view: finch epiblast cells at oviposition likely contain sufficient numbers of naive state cells such that MEK inhibition is able to then stabilize and select for them. The situation in avians is less clear: culture conditions and exogenously introduced genes can significantly alter the pluripotency state of these cells. Transcriptional profiling suggests that chick ES cells, that are cultured in a complex but defined mix of growth factors and additives, are more similar to mouse ES cells than to mouse EpiSC, but are less similar to the chicken blastodermal cells from which they are derived (*Jean et al., 2015*). Chick iPS cells also display characteristics of ES cells, with AP activity detectable in colonies in the presence of MEK inhibition. However, these observations do not suggest how induced

pluripotency relates to normal early embryonic development. Our data suggest that in a minimal media, naive pluripotency in finch blastodermal cells is insensitive to MEK inhibition, whereas chick blastodermal cells require MEK for the expression of pluripotent markers, in these conditions. The challenge is to now further stabilize the naive pluripotent fate of finch blastodermal cells and in combination with recent advances in avian cell culture (*Dai et al., 2014*; *Jean et al., 2015*), develop this technology for understanding and manipulating avian development.

## Materials and methods

### Zebra finch husbandry, eggs and embryos collection

Zebra finches were maintained under a 13:11-hr light–dark cycle with constant room temperature (RT) at 23°C. Eggs were collected every 2 hr from the onset of the light cycle and could be stored at 17–20°C for up to 5 days without any observable drop in hatchability. Sexing of the finches was performed by PCR amplification from genomic DNA isolated from feather pulps (*Bello et al., 2001*). Detailed finch husbandry protocols are available in a separate manuscript (*Mak et al., 2015*).

Fertile chicken eggs were purchased from Shiroyama Farm (Kanagawa, Japan). Staging of the pre-primitive streak embryos followed a system previously described in the chick and are stated as Eyal-Giladi and Kochav stages (EGK) -I to EGK-XIV in this manuscript (*Eyal-Giladi and Kochav, 1976*). Hamburger and Hamilton (HH) system was used for embryos of streak stages and onward (*Hamburger and Hamilton, 1951*). Chicken blastoderms were collected with a paper ring. Finch blastoderms were micro-dissected from eggs by tungsten needles owing to their small size and yolk removed as described below.

### Blastodermal culture

Both chick and finch blastoderms were cut out in Ringer's solution. At early stages, yolk is strongly adherent to the embryo, and the bulk of this was removed by gentle aspiration with a P1000 pipetteman. This left a layer of yolk adherent to the embryo. To remove this layer, the partially cleaned embryos were transferred to 0.45% D-Glucose (Sigma, St. Louis, MO, United States) in HBSS+ (HBSSg) solution and left at RT for 2 hr. After incubation, this yolk layer could be lifted from the embryo by gentle aspiration around the periphery the embryo. Embryos were then be collected and centrifuged at 400×*g* for 2 min. The pellet was resuspended in ED medium (ESGRO [EMD Millipore, Billerica, MA, United States]: DMEM in 1:1) supplemented with Glutamax (Gibco, Waltham, MA., United States), essential amino acid (Gibco), 0.3 mM 2-mercapthoethanol, sodium pyruvate (1:100, Gibco), and 1% fetal calf serum. The cells were plated and grown at different culture conditions. In experiments using MEK inhibitor (PD0325901, Stemgent, Lexington, MA, United States), GSK3ß inhibitors (CHIR99021, Stemgent), the PKC inhibitor (Gö6983 Sigma), and human LIF (LIF1005, EMD Millipore), half to one finch blastoderm per well of an 8-well slide chamber (177445, Nunc Nalgene, Waltham, MA, United States) was used in all experiments unless specified. 1/8 of the chick blastoderm at EGK-X was seeded in each well of an 8-well slide chamber. The growth of avian cells on a variety of substrates was also tested. Substrates tested were 0.1% or 0.25% gelatin (G1393, Sigma) in PBS, collagen type I (C2249, Sigma), type IV (C0543, Sigma), type VI (F1141, Sigma), recombinant E-cadherin (MAB1838, R&D Systems, Minneapolis, MN, United States), growth factor-reduced matrigel (354230, BD Biosciences, Franklin Lakes, NJ, United States), fibronectin (F5022, Sigma) and pronectin F (*Nishishita et al., 2012*) (a kind gift from Shin Kawamata, Foundation of Biomedical Research and Innovation, Kobe, Japan) or laminin (354458, BD Biosciences). Chicken cells grew much more robustly than finch cells and could grow on many different surface types (0.1% or 0.25% gelatin in PBS, collagen type I, type IV, type VI, recombinant E-cadherin, matrigel, fibronectin, pronectin F, or laminin). Optimal growth for chicken blastoderm cells was seen with type I and IV collagen, on which ES colony-like cell aggregates arose after 4-day culture. Optimal growth for finch blastoderm cells was obtained on human placenta type IV collagen-coated surface. We therefore used this coating for all subsequent studies. For Q-PCR analysis, cells were dissociated from the dish using 0.25% Trypsin-EDTA, spun down, and resuspended in lysis buffer before RNA extraction using the RNeasy Micro Kit (Qiagen, Venlo, Netherlands).

## Endogenous AP activity detection and antibody staining of embryos and cultured cells

AP activity was detected by incubating 4% PFA fixed sections of embryos/cultured cells with 4.5 µl/ml NBT (BD Bioscience) and 3.5 µl/ml BCIP (BD Bioscience) substrates. Immunofluorescent staining was performed using standard protocols. Briefly, sections were blocked with 2% skim milk/PBST (0.1% Triton X-100 [Sigma] in PBS) for 1 hr at RT. The primary antibody was applied on 12-to 14-µm cryosections and the slides were incubated overnight at 4°C. Primary antibodies used were: SSEA-1 and EMA-1 (used in 1:200, Developmental studies hybridoma bank), SOX3 (*Wilson et al., 2001*) (in 1:300, a gift of Sara Wilson, Umeå University, Sweden), GATA6 (1:1000, R&D), E-cadherin (1:200, BD Bioscience), and ß-catenin (1:200, BD Bioscience). Slides were rinsed in PBST and appropriate secondary antibodies conjugated with Alexa dyes were applied. The slides were incubated at RT for 4 hr. After rinsing with PBST, the nuclei of the samples were counter-stained with DAPI (1:4000, Dojindo) and mounted in Prolong Gold anti-fade reagent (Invitrogen, Waltham MA, United States). Detection for the finch Vasa homologue was also attempted. Six antibodies against the chick VASA homologue (CVH/DDX) were tested, namely VN1, VN2, VC3, and VC4 (gifts from Bertrand Pain, unpublished); anti-VASA (*Tsunekawa et al., 2000*) and anti-CVH (*Lambeth et al., 2013*). These were used on the sections of zebra finch blastoderms and day 9 gonads, as well as the chicken day 12 gonads as a positive control. All antibodies showed good staining in positive control chicken gonad sections; however, neither finch blastoderm nor finch gonadal section showed immunoreactivity.

## In situ hybridization of whole-mount embryos and cultured cells

In situ hybridization of embryos was performed using standard protocol (*Freter et al., 2008*; *Alev et al., 2013*). Zebra finch-specific and chicken-specific in situ riboprobe templates were generated against cDNA libraries generated by reverse transcription PCR (SuperScript III First-strand synthesis system, Invitrogen) from embryos of EGK-VIII, EGK-X, HH stage 4, HH8 and HH18 (*Supplementary file 1A*).

## Quantitative RT-PCR

Total RNA was isolated from either nine stage-matched newly laid or aged blastoderms of finch or two unincubated blastoderms of chick (EGK-X) (triplicate for each group), using QIAshredder spin columns and RNeasy Micro Kit (Qiagen) according to manufacturer's instructions. First-strand cDNAs were synthesized in parallel from equal amounts of total RNA, using oligo(dT) primers and Superscript III (Invitrogen). Real-time q-PCR was performed in quadruplet using Mesa Green qPCR Mastermix Plus for SYBR Assay (Eurogentec, Seraing, Belgium) and ABI Prism 7900HT Fast Real-Time PCR system (Applied Biosystems, Waltham MA, United States). Melting curve analysis was performed with SDS Software (Applied Biosystems) and gene expression calculated using mean cycle threshold (Ct) values, with normalization to finch or chick glyceraldehyde-3-phosphate dehydrogenase (GAPDH), respectively. Pair-wise comparisons were tested using a t-test. Primer sequences are shown in *Supplementary file 1B*.

## Acknowledgements

We would like to thank Drs Hiroshi Nagashima, Jennifer Nichols, Hitoshi Niwa, Austin Smith, Martin Leeb, Martin Jakt and Masatoshi Takeichi and Ms Kanako Ota for helpful discussions. We are grateful Dr Bertrand Pain for providing various chicken-specific CVH antibodies and Dr Yukiko Nakaya for the E-cadherin staining. We appreciate the help from the Optical image analysis unit at the CDB and the members of the Laboratory for Sensory Development. This work was supported by RIKEN CDB intramural funding for RKL and GS as well as a JSPS fellowship for S-SM, Grant-in-aid for scientific research C from MEXT and director's fund from RIKEN CDB for S-SM and RKL.

## Additional information

### Funding

| Funder | Author |
| --- | --- |
| Japan Society for the Promotion of Science (JSPS) | Siu-Shan Mak |

| Funder | Author |
| --- | --- |
| Ministry of Education, Culture, Sports, Science, and Technology (MEXT) | Siu-Shan Mak, Anna Wrabel, Akira Honda |

The funders had no role in study design, data collection and interpretation, or the decision to submit the work for publication.

## Author contributions

S-SM, RKL, Conception and design, Acquisition of data, Analysis and interpretation of data, Drafting or revising the article, Contributed unpublished essential data or reagents; CA, Acquisition of data, Analysis and interpretation of data; HN, AW, Acquisition of data, Contributed unpublished essential data or reagents; YM, AH, Acquisition of data; GS, Conception and design, Analysis and interpretation of data, Drafting or revising the article

# Additional files

## Supplementary file

• Supplementary file 1. (**A**) Details of primers that were used for generating in situ hybridization riboprobes for the zebra finch. (**B**) Details of primers that were used for quantitative PCR.

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
