## [Decision Letter]

Thank you for sending your work entitled “Characterisation of the finch embryo supports evolutionary conservation of the naïve stage of development in amniotes” for consideration at *eLife*. Your article has been favorably evaluated by Janet Rossant (Senior Editor), Alejandro Sánchez Alvarado (Reviewing Editor), and three reviewers.

The Reviewing Editor and the reviewers discussed their comments before we reached this decision, and the Reviewing editor has assembled the following comments to help you prepare a revised submission.

Mak and colleagues characterize the primitive blastoderm and pluripotent cells of the early zebra finch embryo. They show that oviposition occurs at an earlier developmental stage than the chick allowing access to the potential naïve state of pluripotency. Their data on expression analysis of naïve and primed pluripotency markers and some culture experiments broadly support their claim that the finch blastoderm is in a more naïve state than the chick blastoderm. All reviewers generally considered the work to be an exciting study that explores a novel model organism not previously used to study pluripotency and which potentially allows the study in a bird of a developmental stage likely to correspond to the late blastocyst of the mouse.

There is, however, a general consensus that the work would benefit from a more extensive consideration of the available literature and further characterization of the lines described in Figures 5 and 6. More precisely:

1) There are some recent papers in the literature on generating induced pluripotent stem cells in different avian species and mice. The authors mention nothing about these papers, and how they interpret such their findings in light of these past findings. For example, Rossello et al. (*eLife* 2012) generated and compared iPSC-like cells from finch, chicken, quail, and mice. They claim that the chicken iPSC-like cells are more like the mouse than the finch is, which is the opposite conclusion of the current study in terms of stem cell phenotype. Other studies on avian vs. mouse cells are in [15] (PMID: 25610469), Lu et al. (PMID: 21970437), and many studies from the Pain lab on avian stem cells states, such as the review by [27] (PMID: 23278808). These studies used some of the same genes that Mak et al. used. In addition, the reviewers noted that a more balanced Discussion could be written and should consider [41] “Isolation of Epiblast Stem Cells from Preimplantation Mouse Embryos” in Cell Stem Cell. This paper shows that EpiSCs can be easily derived from pre-epithelialised mouse blastocysts, and this derivation process does not show the strain-dependency of ESC derivation.

2) The data in Figures 5 and 6 are not fully supportive of the finch blastoderm being in a naïve state at oviposition, and the chick being in a primed state. The authors should carry out the necessary statistical analyses to determine whether or not the genes expression differences observed are real or not. A key question that needs to be explicitly addressed by the authors is whether the gene expression pattern of the finch blastoderm becomes more epiblast-like (i.e., more chick-like) with post-oviposition incubation to stage EGK-X? In support of the chick blastoderm containing some naïve cells, Figure 6 appears to have some *AP+ve* and *Nanog+ve* aggregates in the chick panels, although the legend and text state otherwise. Mak and colleagues generated colonies from both types of embryo, but the analyses remain too superficial to support the claim that naïve pluripotent cells can be extracted from pre-epithelial avian epiblast. Therefore, colonies formed from explanting single embryo cells should be characterised in more detail utilizing additional markers of naïve pluripotency, such as those shown in Figure 5.

On the whole, this is a very interesting and well-presented study providing the groundwork for more extensive characterization of the putative naïve state in avian embryos. However, further work is required before the conservation of the naïve state of pluripotency between mouse and bird can be concluded.

---

## [Author Response]

*Mak and colleagues characterize the primitive blastoderm and pluripotent cells of the early zebra finch embryo. They show that oviposition occurs at an earlier developmental stage than the chick allowing access to the potential naïve state of pluripotency. Their data on expression analysis of naïve and primed pluripotency markers and some culture experiments broadly support their claim that the finch blastoderm is in a more naïve state than the chick blastoderm. All reviewers generally considered the work to be an exciting study that explores a novel model organism not previously used to study pluripotency and which potentially allows the study in a bird of a developmental stage likely to correspond to the late blastocyst of the mouse*.

*There is, however, a general consensus that the work would benefit from a more extensive consideration of the available literature and further characterization of the lines described in*
Figures 5 and 6*. More precisely:*

As detailed below we have performed this additional work, which supports our original contention that the finch blastoderm at oviposition corresponds to the late blastocyst stage of the mouse, and contain naïve pluripotent cells.

*1) There are some recent papers in the literature on generating induced pluripotent stem cells in different avian species and mice. The authors mention nothing about these papers, and how they interpret such their findings in light of these past findings. For example, Rossello et al. (*eLife *2012) generated and compared iPSC-like cells from finch, chicken, quail, and mice. They claim that the chicken iPSC-like cells are more like the mouse than the finch is, which is the opposite conclusion of the current study in terms of stem cell phenotype. Other studies on avian vs. mouse cells are in*
[15]
*(PMID: 25610469), Lu et al. (PMID: 21970437), and many studies from the Pain lab on avian stem cells states, such as the review by*
[27]
*(PMID: 23278808). These studies used some of the same genes that Mak et al. used. In addition, the reviewers noted that a more balanced Discussion could be written and should consider*
[41]
*“Isolation of Epiblast Stem Cells from Preimplantation Mouse Embryos” in Cell Stem Cell. This paper shows that EpiSCs can be easily derived from pre-epithelialised mouse blastocysts, and this derivation process does not show the strain-dependency of ESC derivation*.

These studies are now discussed and new references are added. The Discussion is re-written in light of them. Our main purpose is to understand how normal development relates to the derivation of ES cells. The studies from both the Pain and Jarvis labs do suggest that the chick ES and chick IPS cells exhibit certain features similar to mouse ES cells. However, in both cases significant manipulation is involved – in the case of the Pain group by manipulating culture conditions, and in the case of the Jarvis group, introducing the Yamanaka factors. In this work, we used direct comparison (between chick and finch embryos, and between early finch and late finch embryos) without any experimental manipulation, or used mouse ES cell culture conditions and investigated the effect of only the MEK inhibitor and LIF without any other inhibitor(s). This enabled a more straightforward comparison. Our data suggest that finch embryos at oviposition exhibit clear molecular and morphological features indicative of them being at a naïve state of development, whereas chick embryos show features of them being closer to the primed state. That said, we do not think that data from our lab and other labs contradict each other for two reasons. First, although chick embryos are at a primed developmental state, the molecular data also suggested that they are less primed than originally assumed, as it retains some levels of expression of naïve pluripotent markers. Second, we do not think that there is a fundamental difference between the chick and finch embryos, aside from their ovipositional stages; and all of these studies combined support our claim that there is a naïve state of development in birds and conservation of the naïve state of embryogenesis amongst amniotes.

*2) The data in*
Figures 5 and 6
*are not fully supportive of the finch blastoderm being in a naïve state at oviposition, and the chick being in a primed state. The authors should carry out the necessary statistical analyses to determine whether or not the genes expression differences observed are real or not. A key question that needs to be explicitly addressed by the authors is whether the gene expression pattern of the finch blastoderm becomes more epiblast-like (i.e., more chick-like) with post-oviposition incubation to stage EGK-X? In support of the chick blastoderm containing some naïve cells,*
Figure 6
*appears to have some* AP+ve *and* Nanog+ve *aggregates in the chick panels, although the legend and text state otherwise. Mak and colleagues generated colonies from both types of embryo, but the analyses remain too superficial to support the claim that naïve pluripotent cells can be extracted from pre-epithelial avian epiblast. Therefore, colonies formed from explanting single embryo cells should be characterised in more detail utilizing additional markers of naïve pluripotency, such as those shown in*
Figure 5.

We thank the reviewers for these comments. As the reviewers had suggested, we performed additional experiments by comparing ovipositional finch embryos before and after incubation (i.e., comparing EGK-VI-VIII finch embryos with EGK-X-XI finch embryo). The data (presented in revision Figure 5) is consistent with our chick/finch comparison (revision Figure 4). In both types of comparisons, no cDNA amplification was involved, and multiple biological replicates were prepared and quadruple Q-PCR reactions for each sample were performed. We think our Q-PCR data are statistical rigorous. We do agree with the reviewers that cross-species comparison of Q-PCR data should be interpreted with caution. We therefore would like to thank the reviewers for suggesting this experiment, which is now shown in revised Figure 5, which provided even stronger evidence for a transition from naïve to primed stages as development proceeds. We also tested the other experiment suggested by this reviewer, by performing Q-PCR analysis of cultured blastoderm cells (revision Figure 6). Due to the small amounts of RNA obtainable from a single colony and subsequent errors that could be introduced by cDNA amplification, as well as the “user bias”, all cells within a single well were used for Q-PCR analysis. Molecular data from this experiment suggest that MEK inhibition helps retain naïve markers in cultured blastoderm cells (revision Figure 6). The results together strengthen our conclusions.